METHODS

# Multimodal CustOmics: A unified and interpretable multi-task deep learning framework for multimodal integrative data analysis in oncology

**Hakim Benkirane**[1,2,3,4], **Maria Vakalopoulou**[1,2], **David Planchard**[4,5], **Julien Adam**[4,6], **Ken Olaussen**[4,7], **Stefan Michiels**[3,8☯], **Paul-Henry Cournède**[1,2☯]*

**1** Université Paris-Saclay, CentraleSupélec, Laboratory of Mathematics and Computer Science (MICS), Gif-sur-Yvette, France, **2** IHU PRISM, National PRecISion Medicine Center in Oncology, Gustave Roussy, Villejuif, France, **3** Inserm U1018 Oncostat, Équipe Labellisée Ligue Contre le Cancer, Université Paris-Saclay, Villejuif, France, **4** Université Paris-Saclay, Faculty of Medicine, Le Kremlin Bicêtre, France, **5** Department of Medical Oncology, Thoracic Group and International Center for Thoracic Cancers (CICT), Gustave Roussy, Université Paris-Saclay, Villejuif, France, **6** Inserm U1186, Gustave Roussy, Université Paris-Saclay, Villejuif, France, **7** Inserm U981, Gustave Roussy, Université Paris-Saclay, Villejuif, France, **8** Bureau de Biostatistique et d'Épidémiologie, Gustave Roussy, Université Paris-Saclay, Villejuif, France

☯ These authors contributed equally to this work.
* paul-henry.cournede@centralesupelec.fr

**Data availability statement:** The datasets analyzed in this study are publicly available

## Abstract

Characterizing cancer presents a delicate challenge as it involves deciphering complex biological interactions within the tumor's microenvironment. Clinical trials often provide histology images and molecular profiling of tumors, which can help understand these interactions. Despite recent advances in representing multimodal data for weakly super-vised tasks in the medical domain, achieving a coherent and interpretable fusion of whole slide images and multi-omics data is still a challenge. Each modality operates at distinct biological levels, introducing substantial correlations between and within data sources. In response to these challenges, we propose a novel deep-learning-based approach designed to represent multi-omics & histopathology data for precision medicine in a readily interpretable manner. While our approach demonstrates superior performance compared to state-of-the-art methods across multiple test cases, it also deals with incomplete and missing data in a robust manner. It extracts various scores characterizing the activity of each modality and their interactions at the pathway and gene levels. The strength of our method lies in its capacity to unravel pathway activation through multimodal relationships and to extend enrichment analysis to spatial data for supervised tasks. We showcase its predictive capacity and interpretation scores by extensively exploring multiple TCGA datasets and validation cohorts. The method opens new perspectives in understanding the complex relationships between multimodal pathological genomic data in different cancer types and is publicly available on Github.

from the NIH Genomic Data Commons Data Portal (https://gdc.cancer.gov/about-data/publications/pancanatlas). The code for the Multimodal CustOmics model, including preprocessing pipelines, model architecture, and training scripts, is available at: https://github.com/HakimBenkirane/Multimodal-CustOmics.

**Funding:** This work was supported by the Prism project, funded by the Agence Nationale de la Recherche under grant number ANR-23-IAHU-0002 (HB, MV, PHC) and by the Public Health graduate school of Paris-Saclay University (HB). The funders had no role in the study design, data collection and analysis, decision to publish, or preparation of the manuscript.

## Author summary

Cancer diagnosis and treatment require a deep understanding of the intricate biological interactions within tumors. In this study, we developed Multimodal CustOmics, a deep-learning framework designed to integrate histopathology images and multi-omics data, providing a comprehensive approach to cancer characterization. Multimodal CustOmics excels in predictive performance for both classification and survival analysis, outperforming current state of the art methods. Beyond its predictive capabilities, Multimodal CustOmics offers multi-level interpretability, revealing critical insights into genes, pathways, and spatial interactions. This interpretability is essential for clinicians and researchers to understand the underlying mechanisms of cancer progression and treatment responses. Our model's robustness to missing data ensures its feasibility in real-world clinical scenarios. By leveraging diverse datasets from TCGA, Multimodal CustOmics has shown its potential to offer new avenues in precision medicine, and highlights the importance of integrating diverse biological data to achieve a holistic understanding.

## 1. Introduction

The rapid advancements in deep learning have significantly impacted the medical field, especially in tasks such as cancer diagnosis, prognosis, and therapeutic response prediction. These tasks often necessitate a comprehensive characterization of a patient's profile, which involves integrating heterogeneous data sources, including histopathology slides, molecular profiles, and clinical data [5,19–21,39,43,48,52].

Histopathology slides, or Whole-Slide Images (WSIs), are the gold standard in computational pathology for prediction tasks. These gigapixel images provide detailed visual information about tissue architecture and cellular morphology, making them invaluable for diagnosing various diseases, including cancer [17]. However, the high resolution of WSIs poses significant computational challenges, particularly in supervised learning tasks, which require efficient training and robust data representations. Current approaches, such as Multiple Instance Learning (MIL), extract small patches from WSIs and process them independently before aggregating their representations. Despite their utility, these methods often fail to capture the spatial context and heterogeneity of the tumor microenvironment [36,40].

Simultaneously, multi-omics data—including Copy Number Variations (CNV), RNA sequencing (RNAseq), and DNA methylation profiles—provide a molecular-level understanding of the biological processes underlying diseases. Integrating these diverse data types can reduce uncertainties arising from different experimental conditions and reveal complex biological interactions not apparent from single data sources [1]. Nevertheless, the high dimensionality and heterogeneity of multi-omics data present significant challenges. Each molecular source may encapsulate distinct biological functions, necessitating sophisticated methods to extract clinically relevant insights.

Integrating WSIs and multi-omics data is a promising yet challenging area in computational pathology. Combining the rich visual information from histopathological images with molecular profiles can enhance diagnostic accuracy, prognostic predictions, and therapeutic decisions. Despite notable efforts in this domain, such as those by Courtiol et al. [53] and Lu et al. [54], current methods often fail to leverage the full potential of multimodal data. For instance, Courtiol et al. combined deep features from histology images with transcriptomic

data, while Lu et al. integrated genomic and histopathological data using a unified deep learning architecture. However, these approaches often rely on early integration techniques or standard feed-forward networks, which may not fully exploit the rich information provided by multi-omics data.

Moreover, a critical challenge in multimodal integration is dealing with incomplete datasets, a common issue in clinical settings where one or more modalities may be missing. Addressing this challenge is essential for the practical application of multimodal integration techniques.

This work introduces a novel method, Multimodal CustOmics, designed to integrate histology images with the patient's molecular profile effectively. Our approach captures high correlations between and within different dimensions of the biological system, such as biological pathways or cellular regions. Utilizing a multiple-instance variational autoencoder with modality dropout, our method integrates multimodal data while remaining robust to missing data. We adapt the autoencoder to create a mixture-of-experts representation, similar to the approach by Jordan et al. [24], which enhances interpretability by aligning multiple relevant biological dimensions of the patient's profile.

**Our method provides three significant advantages:**

- **Enhanced Understanding:** By dividing each modality into different functional groups, we achieve a more nuanced understanding of high-dimensional problems.
- **Robustness to Missing Data:** Implementing modality dropout within our framework creates an end-to-end pipeline resilient to missing modalities.
- **Interpretable Interactions:** The multidimensional representation facilitates easy interpretation of interactions between all dimensions from different modalities, thanks to the mixture-of-experts networks.

## 2. Materials and methods

To tackle the challenges of integrating multimodal data, we propose a deep-learning framework to create an interpretable image-omic representation, represented in Fig 1, that captures interactions at multiple levels of the biological system. This integration method is named Multimodal CustOmics and builds upon the strategy introduced in our previous work [3]. The original network optimally integrated heterogeneous data from different omics sources while preserving the specificity of each modality. The new version of our multimodal network can jointly integrate H&E slides and molecular profile features (mutation status, copy-number variation, RNA sequencing [RNA-seq] expression, DNA methylation, etc.). Additionally, it can interpret how the interaction between all those sources correlates with specific supervised tasks such as molecular subtype identification or survival prediction. Furthermore, this method facilitates the assessment of feature importance at multiple levels through ad-hoc scores computed within the framework: gene level, pathway level, and spatial level.

At the gene level, the method outputs the importance score of each gene for each molecular source, both independently and in association with other modalities. At the pathway level, a Multimodal Pathway Enrichment Score (MPES) is computed to assess the importance of a specific pathway for a specific prediction task, such as molecular subtype classification or survival prediction. This score has also been extended to account for spatial correlations in the histopathology slide and reflects the importance of the interaction between spatial regions of the WSI and pathways.

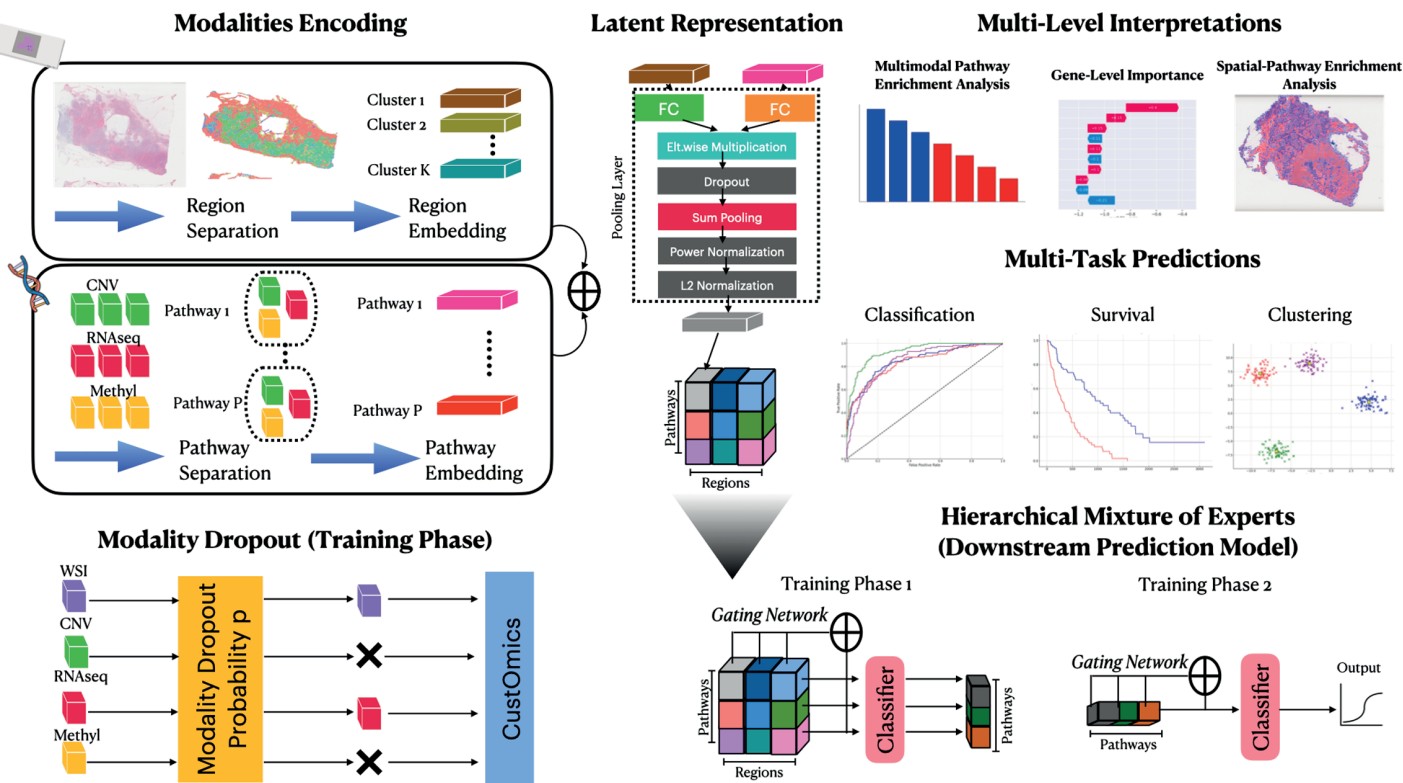

**Fig 1. Modality encoding.** Each modality undergoes a specific methodology. For histopathology slides, spatial regions are extracted using a hypergraph encoder to obtain an embedding for each region. Genes are segregated into gene sets in multi-omic integration, resulting in a multi-omics embedding per set. **Modality Dropout:** Dropout Layer for modalities to deal with missing modalities by relaxing the constraint of needing all modalities at once. **Latent Representation:** The latent representation comprises multiple blocks, each representing the embedding of the interaction between a region and a pathway. **Hierarchical Mixture of Experts:** Prediction Model based on MoE architecture on the different block embeddings. Phase 1 learns the weights of each region with regards to a fixed pathway while phase 2 learns the weights of each pathway for the final prediction. **Multi-Task Predictions:** The latent representation is then utilized for supervised tasks such as classification or survival analysis or unsupervised tasks for tasks like clustering. **Multi-Level Interpretations:** Interpretation results are extracted at various levels: gene, gene-set, and spatial levels.

Additionally, to further explain spatial interpretability results, we extract high-attention image patches and analyze them using the pre-trained Hover-Net model for cell instance segmentation and classification [13]. This analysis categorized cells into tumors, lymphocytes, stromal, necrosis, and epithelial cells. We quantitatively assessed the frequency of these cell types in high-attention patches for each patient. Additionally, we estimated the proportion of Tumor Infiltrating Lymphocytes (TILs) in these patches using the methodology developed by Saltz et al. [35], thus providing a deeper understanding of the tumor microenvironment.

## 2.1. Model construction

Within the scope of this study, we design, implement, and evaluate a multimodal integration network for integrating histopathology slides and multi-omics data. For $1 \leq i \leq N$, let us denote by $W_i$ and $O_i$, respectively, the WSI and multi-omics bag for patient $i$. The goal of this study is to build and train a multi-task network $\mathcal{M}$ that takes as input the two bags and creates an interpretable multimodal representation $z_i$ for each patient such that $\mathcal{M}(W_i, O_i) = z_i$.

**2.1.1. Representation of histopathology images.** To effectively represent histopathology slides, we opted for a hypergraph approach to address the complex relationships within

Whole Slide Images (WSIs) that cannot be effectively captured by traditional graph-based methods. Unlike standard graphs, which are limited to pairwise relationships, hypergraphs can model higher-order interactions among multiple entities, such as patches within a WSI. This is particularly relevant in histopathological analysis, where regions of interest often exhibit intricate morphological and spatial interdependencies. By leveraging a hypergraph structure, our method enables a more nuanced representation of these interactions, offering enhanced interpretability and adaptability in multimodal settings. Additionally, the hypergraph framework introduces computational efficiency compared to transformer-based architectures by maintaining a lower parameter count, aligning with the goal of dividing the multimodal representation learning task into smaller, interpretable strata as envisioned in Multimodal CustOmics.

To this end, an automated tissue segmentation is conducted using the method outlined in Lu et al. [30], which uses Otsu Binarization to separate the background from the tissue. Subsequently, non-overlapping patches of size 256x256 are extracted at a magnification of 20x. These patches are then given as input to a ResNet-18, trained with a contrastive strategy on histopathology data akin to Ciga et al. [10], generating a 1024-dimensional feature vector $h \in \mathbb{R}^{1024}$ for each patch. The set of features $(h_j)_{1 \le j \le n_p}$ associated with a set $W_i$ comprising $n_p$ patches is consolidated into a feature matrix $\mathbf{X}_i \in \mathbb{R}^{n_p \times 1024}$. Each patch $x_j$ is defined by its ResNet-18 feature representation $h_j$, encapsulating morphological properties, and a set of coordinates $g_j = (g_{x,j}, g_{y,j})$ representing the spatial center of the patch.

Following the methodology of Benkirane et al. [4], a patch filtering process is executed to summarize information in the Whole Slide Image (WSI). This involves agglomerative clustering, where a similarity kernel is computed between all patch pairs. Two similarity matrices, $K_h \in \mathbb{R}^{n_p \times n_p}$ and $K_g \in \mathbb{R}^{n_p \times n_p}$ are computed, where $K_h = (\kappa_h(x_i, x_j))_{1 \le i,j \le n_p}$ and $K_g = (\kappa_g(x_i, x_j))_{1 \le i,j \le n_p}$. Here, $\kappa_h(x_i, x_j) = e^{-\lambda_h \|h_i - h_j\|^2}$ represents a morphological similarity metric, and $\kappa_g(x_i, x_j) = e^{-\lambda_g \|g_i - g_j\|^2}$ signifies a spatial proximity metric. The overall similarity kernel $\kappa(x_i, x_j) = \kappa_h(h_i, h_j)\kappa_g(g_i, g_j)$ is used to measure the similarity between patches by incorporating both histopathological ($\kappa_h$) and genomic ($\kappa_g$) feature similarities. To cluster highly similar patches, we apply agglomerative clustering, where patches $x_i$ and $x_j$ are assigned to the same cluster $C_k$ if $\kappa(x_i, x_j) \ge \delta_c$. This ensures that only patches with a similarity above the threshold $\delta_c z$ are merged, preserving structurally and functionally coherent regions. These clusters are then aggregated into a single representation $p_k = (\tilde{h}_k, \tilde{g}_k)$, where $\tilde{h}_k = \frac{1}{\|C_k\|} \sum_{j \in C_k} h_j$ and $\tilde{g}_k = \frac{1}{\|C_k\|} \sum_{j \in C_k} g_j$. This new patch representation approximates a localized region, effectively reducing the number of instances in WSIs while preserving a substantial amount of information.

To identify multiple regions of interest within the Whole Slide Image (WSI), we begin by dividing the extracted set of patches, denoted as $\mathcal{P}_i$, into $K$ distinct clusters using the K-Means clustering algorithm. Each of these clusters represents a different category of regions within the WSI.

Once the clustering is completed, we define the neighborhood relationship between patches. Specifically, for each patch $p_j \in \mathcal{P}_i$, we define its neighborhood as:

$$\gamma(p_j) = \{p_k \in \mathcal{P}_i \mid \kappa_h(p_k, p_j) \ge \delta_h\}$$

where:

- $\kappa_h(p_k, p_j)$ is a similarity function that quantifies the relationship between patches $p_k$ and $p_j$.

- $\delta_h$ is a predefined threshold that determines whether two patches should be considered neighbors.

Using this neighborhood definition, we proceed to construct a hypergraph. The hypergraph structure is represented by an incidence matrix $\mathbf{H} \in \mathbb{R}^{|\mathcal{P}_i| \times |E_i|}$, where $|\mathcal{P}_i|$ is the total number of patches, and $|E_i|$ is the number of hyperedges. Each entry $h(k,j)$ in the matrix is defined as follows:

$$h(k,j) = \begin{cases} 1, & \text{if } p_j \in \gamma(p_k) \\ 0, & \text{otherwise} \end{cases}$$

Thus, the hyperedges in the hypergraph are formed based on patch similarity, and the incidence matrix $\mathbf{H}$ encodes these relationships, indicating which patches belong to which hyperedges.

Next, the hypergraph is processed using a Graph Neural Network (GNN) that applies a series of hypergraph convolutional layers and attention mechanisms [2]. This GNN extracts meaningful representations for each patch, resulting in a feature matrix:

$$\mathbf{X}_i^{GNN} \in \mathbb{R}^{|\mathcal{P}_i| \times z_{dim}}$$

where each row corresponds to a learned feature vector of dimension $z_{dim}$ for a patch in $\mathcal{P}_i$.

To obtain a region-level representation, we aggregate the patch-level features into region embeddings. Specifically, patches belonging to the same region (as defined by the initial K-Means clustering) are pooled together to form a compact feature representation for that region:

$$\mathbf{X}_i^K \in \mathbb{R}^{K \times z_{dim}}$$

where each row of $\mathbf{X}_i^K$ corresponds to the embedding of one of the $K$ regions in the WSI. This final region-level representation serves as the condensed representation of the entire WSI, capturing spatial and contextual relationships across different regions.

**2.1.2. Integration of multi-omics data.** We consider $S$ omic sources in the set $O_i = (O_{i,s})_{1 \le s \le S}$. The initial step in processing these omic sources involves partitioning each source into $P$ gene sets, each possessing distinct functional properties, defined as $O_{i,s} = (O_{i,s,p})_{1 \le p \le P}$. Subsequently, we engage in Variational Autoencoder-based representation learning for each of these $P$ distinct gene sets, a concept introduced in [3]. The encoding networks $(\mathcal{C}_p)_{1 \le p \le P}$ are designed to accept inputs from all omic sources about each gene set, encapsulated by the operation $\mathcal{C}_p(O_{i,1,p}, \dots O_{i,S,p}) = z_{i,p}^O \in \mathbb{R}^{z_{dim}}$. These representations are then concatenated to form a final matrix $\mathbf{O}_i^P \in \mathbb{R}^{P \times z_{dim}}$, serving as the comprehensive collection of multi-omics data.

**2.1.3. Multimodal representation and prediction.** Upon acquiring the WSI bag $\mathbf{X}_i^K$ and the multi-omics bag $\mathbf{O}_i^P$ for a given patient $i$, the next step involves constructing a joint multimodal representation that effectively integrates spatial and molecular features. This representation is obtained through a bilinear operation applied to each pair of elements from both modalities, generating a three-dimensional tensor:

$$\mathbf{Z}_i \in \mathbb{R}^{K \times P \times z_{dim}} \tag{1}$$

where each entry is computed as:

$$z_{i,k,p} = \mathcal{B}((\mathbf{X}_i^K)_k, (\mathbf{O}_i^P)_p) \tag{2}$$

with $\mathcal{B}$ denoting the bilinear fusion operator. This operator is adapted from the multimodal factorized bilinear pooling (MFB) method introduced by Yu et al. [46], which enables efficient feature fusion by capturing second-order interactions between modalities while reducing computational complexity compared to traditional cross-attention mechanisms. By leveraging this approach, the model preserves the informative relationships between the histopathological and molecular features in a structured manner. The resulting multimodal representation $\mathbf{Z}_i$ is then processed by a hierarchical mixture of experts (MoE) network [24], designed to model pathway-specific and global patient-level interactions. The first stage of this MoE network focuses on pathway-specific feature learning by computing a pathway representation $z_{i,p}^{moe}$ for each pathway $p$ through a weighted aggregation of the associated spatial regions:

$$z_{i,p}^{moe} = \sum_{k=1}^{K} w_{i,k}^{p} z_{i,p,k} \tag{3}$$

where $w_{i,k}^{p}$ represents the learned importance weight assigned to spatial region $k$ within pathway $p$, subject to the constraint $\sum_{k=1}^{K} w_{i,k}^{p} = 1$ to ensure a valid probability distribution. These trainable parameters $(w_{i,k}^{p})_k$ allow the model to focus on the most informative regions relevant to each pathway, effectively prioritizing key spatial features for downstream analysis. Each pathway representation $z_{i,p}^{moe}$ is subsequently passed through a linear transformation to generate pathway-specific predictions. In the second stage, the learned pathway representations are aggregated to form a unified patient-level representation $z_i^{moe}$, computed as:

$$z_i^{moe} = \sum_{p=1}^{P} w_{i,p} z_{i,p}^{moe} \tag{4}$$

where $w_{i,p}$ denotes the learned importance weight assigned to pathway $p$, with $\sum_{p=1}^{P} w_{i,p} = 1$ ensuring a valid probability distribution. The parameters $(w_{i,p})_p$ are optimized during training to emphasize the most relevant pathways for final decision-making. The resulting patient-level representation $z_i^{moe}$ is then passed through a final linear layer, mapping it to the model's final prediction. By structuring the multimodal fusion through hierarchical MoE networks, the proposed framework effectively models both local (pathway-specific) and global (patient-level) interactions, enabling robust integration of spatial and molecular features for improved predictive performance.

**2.1.4. Downstream tasks.** Multimodal CustOmics accommodates training for two distinct tasks. Firstly, a supervised classification task aims to predict the probability of each class occurrence. This task undergoes training with a standard Categorical Cross-Entropy loss computed between the predicted classes and the ground truth labels.

The second task involves predicting survival outcomes, trained using the DeepSurv loss function outlined in Katzman et al. [25]. The model adopts the negative partial log-likelihood formula, expressed in our context as:

$$L(\theta) = -\sum_{i:E_i=1} \left( \hat{\mu}(x_i; \theta) - \log \sum_{j \in \mathcal{R}(T_i)} e^{\hat{\mu}(x_j; \theta)} \right) \tag{5}$$

where $E_i$ represents the event for patient $i$, $\hat{\mu}(x; \theta)$ denotes the risk function associated with the risk score estimated by the network's output layer, and $\mathcal{R}(t)$ defines the risk set, signifying the patients still susceptible to failure after time $t$.

This method can also perform self-supervised clustering based on the DeepCluster framework [9], although this aspect will not be detailed in this paper.

**2.1.5. Multimodal dropout.**   To better enforce the robustness of our method to missing data, we implement multimodal dropout introduced in Cheerla et al. [7] to deal with missing modalities under the assumption of modalities missing at random. Instead of dropping single neurons, the idea is to drop entire feature vectors corresponding to specific modalities so that it scales up the weights of the others. This is applied to each sample data with a probability $p$ for each modality. Dropout is implemented differently across modalities: for omics, entire modalities are randomly masked during training within the variational autoencoders' input layer, while for WSI, dropout occurs before the multimodal fusion network. The dropout rate is a hyperparameter to tune.

## 2.2. Multi-level interpretability

To make the results of the Multimodal CustOmics model interpretable, we implement multiple scores to understand the predictions at different levels of the integration process.

**2.2.1. Gene importance and pathway enrichment.**   In pursuit of enhancing gene-level interpretability, we adapt a method introduced in Withnell et al. [44] to compute SHAP (Shapley Additive Explanations) values for deep variational autoencoders, as described in Lundberg et al. [31]. Extending this methodology to the multimodal setting, akin to our formulation is a critical objective.

After training the Multimodal CustOmics network, the multi-omics embedding part calculates SHAP values for genes or latent dimensions. These values estimate the contribution of each element by averaging the SHAP values across samples with similar features. The computed SHAP values provide insights at various training phases, highlighting gene importance in single-omic, multi-omic, and multimodal integration. Detailed explanations of these processes can be found in Text S1.

Furthermore, aiming for a more biologically interpretable model, we propose the derivation of a Pathway Enrichment Score (PES) to assess the impact of specific pathway activations in prediction tasks. Leveraging the weights learned by the gating networks of the mixture of experts, we define a ranking score $r_{ip}$ per patient and pathway. This score ($r_{ip} = (w_{i,p})$) gauges the overall pathway contribution to the final prediction, forming the basis for pathway ranking for enrichment analysis.

Upon computing these scores, we draw inspiration from Lundberg et al. [18] to conduct gene set variation analysis using the importance scores. For each patient $i$, we denote $s_{i,g}$ as the SHAP value computed by Multimodal CustOmics. To ensure generalization across all pathways, these values are normalized by the pathway importance computed for the MoE representation, resulting in the normalized SHAP value $\tilde{s}_{i,g} = r_{i,p(g)} s_{i,g}$, where $p(g)$ represents the pathway corresponding to the gene $g$.

Subsequently, these normalized scores serve as rankings for computing associated p-values for each pathway. This is achieved by employing a Kolmogorov-Smirnov test, comparing the distribution of genes in the pathway versus the others.

**2.2.2. Multimodal interaction score.**   To evaluate the importance of the interaction between a spatial region and a functional group, we compute a Multimodal Interaction Score (MIS). This score is directly obtained from the weights of the gating network such that for a specific patient $i$, we have $MIS_{i,k,p} = (w_k^p)_i$. The score measures the impact of a multimodal interaction between a spatial region and a functional group on the final prediction.

## 2.3. Datasets

This study uses the pan-cancer dataset from the Genomic Data Commons (GDC) [14], comprising 11,768 patients across 33 tumor types and encompassing multi-omics data, histopathology slides, and clinical data. We assess the performance of our model on both the entire pan-cancer dataset and smaller cohorts of specific tumor types to demonstrate the robustness of our approach concerning varying patient numbers. The objective is to evaluate Multimodal CustOmics for tumor type classification and survival outcome prediction. We selected eight cohorts based on patient numbers and censoring rates for the survival prediction task, as outlined in S1 Table. Three of these eight cohorts were also utilized to assess classification into molecular subtypes: TCGA-BRCA, TCGA-COAD, and TCGA-STAD. To validate the interpretability results provided by our method, we employed a private validation dataset derived from the International Adjuvant Lung Cancer Trial (IALT) [15]. This study included patients with non-small-cell lung cancer, with available histopathology slides, mutation, and copy-number variations data. Further details can be found in S2 Table. The validation entails predicting survival outcomes using histopathology slides and DNAseq data. We compare the results of this validation dataset with those of the TCGA-LUAD and TCGA-LUSC cohorts.

## 2.4. Implementation details

The Multimodal CustOmics framework is based on the Pytorch deep-learning library [34]. It can be applied to any combination of high-dimensional datasets and histopathology images with multitask training. As done in Zhang et al. [49], DNA methylation data can be divided into 23 separate blocks, each feeding a hidden layer corresponding to a chromosome to avoid overfitting and save GPU memory.

Inspired by the work in [3], we adopt a multiphase training strategy to ensure the optimal integration of all modalities. During the first phase, we train each modality independently to obtain unsupervised sub-representations for the different bags. The second phase consists of unfreezing the central encoding network that learns the supervised representation of the final omics bag.

The whole architecture is built using fully connected blocks with weights initialized following a uniform distribution $\mathcal{U}\left(-\frac{1}{\sqrt{k}}, \frac{1}{\sqrt{k}}\right)$ where $k$ is the number of weight parameters. We use a batch normalization technique in each layer composing the neural network to address the internal covariate shift problem [23]. Also, to avoid overfitting problems, we use dropout [38]; its rate is considered a hyperparameter.

The input dataset was randomly split into training, validation, and testing sets (60-20-20%) using stratified 5-fold cross-validation so that the proportion of samples in each tumor type between the different sets is preserved in all the folds. We perform Bayesian optimization [37] using the validation set to find our model's best possible combination of hyperparameters.

For the hypergraph encoding, S1C Fig demonstrates how model performance varies across key hyperparameters, guiding the optimal configuration for initial hypergraph construction.

Regarding omics data, RNA-seq data is represented as normalized expression values, CNV data as segment mean values, and methylation as beta values. All features are mapped to gene-level representations using standard annotations: CNVs via UCSC genome browser coordinates and methylation probes through manufacturer's gene annotations.

## 3. Experimental setup

To evaluate the effectiveness of Multimodal CustOmics, we compare it against four established methods that integrate omics and histopathology data. For this comparison, we follow the study conducted by Chen et al. [8] and use the engineered baselines introduced for omics, histopathology, and multi-omics integration. In particular, we compare with:

- **SNN architecture [27]:** We compare against a feed-forward self-normalizing network (SNN) architecture [27] as a multi-omics baseline, where multi-omics data are concatenated before being fed into the network. This architecture, used in previous works [8, 55], is considered state-of-the-art for histology-genomic integration.
- **DeepSets [47]:** We employ DeepSets, one of the first neural architectures designed for set-based deep-learning problems. DeepSets utilizes sum pooling over instance-level features. Its multimodal extension, presented by Chen et al. [8], processes omics data with an SNN and integrates them using bilinear pooling.
- **Attention MIL [22]:** We consider Attention MIL, a set-based network similar to DeepSets that replaces sum pooling with an attention pooling technique .
- **DeepAttnMISL [45]:** We include DeepAttnMISL, a set-based network that first applies K-Means clustering to instance-level features. Each cluster is processed using Siamese networks, and the cluster features are then aggregated using global attention pooling .
- **MCAT [8]:** We compare our method to MCAT, presented by Chen et al. , a transformer-based set-based network that represents the current state-of-the-art in multimodal histology-genomics integration. MCAT combines modalities using bilinear attention-based pooling.
- **SurvPATH [56]:** We include SurvPath, introduced by Jaume et al., a new transformer-based architecture that embeds transcriptomics information in the form of biological pathways. The omics layer in SurvPATH consists of an SNN architecture adapted to consider multi-omics data.
- **HEALNet [26]:** We also compare against HEALNet, introduced by Hemker et al., a multimodal fusion architecture designed for heterogeneous biomedical data integration. HEALNet preserves modality-specific information, captures cross-modal interactions, and demonstrates robustness to missing modalities. It achieves state-of-the-art performance in multimodal survival analysis across cancer datasets.

We follow each method's experimental protocols outlined in their respective publications to ensure a fair comparison.

## 4. Results

### 4.1. Prediction results

We executed multiple test cases in a comprehensive evaluation of Multimodal CustOmics within a multitask framework encompassing both classification and survival analyses. The performances for the classification and survival tasks are presented in Tables 1 and 2. We assess model performance using two sets of metrics: classification metrics (balanced accuracy, F1-score, precision, recall, and Area Under ROC Curve (AUC)) and survival analysis metrics (concordance-index (C-index) [6]). We initially assessed cancer-type classification within the TCGA Pancancer cohort, revealing Multimodal CustOmics' superior performance in terms of AUC compared to other benchmarked methods [16]. This notable performance can be attributed to the substantial patient cohort size and the rich information embedded within molecular data, a phenomenon well-documented across multiple studies [3,49,50].

**Table 1. Classification performances.** Comparison of the classification performances for the 4 tasks concerning the area under ROC (AUC %) $\pm$ the standard deviation across the folds.

| Methods | PANCAN | BRCA | COAD | STAD |
|---|---|---|---|---|
| SNN [27] (Multi-Omics) | 94.1 $\pm$ 2.7 | 92.0 $\pm$ 3.3 | 79.2 $\pm$ 3.3 | 84.6 $\pm$ 3.3 |
| HealNET (Multi-Omics) | 98.7 $\pm$ 1.5 | 98.0 $\pm$ 1.2 | 87.8 $\pm$ 2.0 | 98.1 $\pm$ 1.3 |
| **CustOmics (Multi-Omics)** | **98.9 $\pm$ 1.4** | **98.3 $\pm$ 1.0** | **88.1 $\pm$ 1.9** | **98.4 $\pm$ 1.2** |
| DeepSets [47] (WSI Only) | 84.7 $\pm$ 3.3 | 68.6 $\pm$ 4.1 | 55.2 $\pm$ 4.3 | 58.9 $\pm$ 4.6 |
| AttnMIL [22] (WSI Only) | 88.4 $\pm$ 2.0 | 71.2 $\pm$ 4.9 | 56.2 $\pm$ 4.4 | 61.4 $\pm$ 4.4 |
| DeepAttnMISL [45] (WSI Only) | 89.8 $\pm$ 2.5 | 71.1 $\pm$ 3.3 | 55.7 $\pm$ 4.0 | 62.1 $\pm$ 4.5 |
| MCAT [8] (WSI Only) | 90.4 $\pm$ 1.8 | 72.3 $\pm$ 3.3 | 61.7 $\pm$ 3.3 | 69.4 $\pm$ 3.5 |
| SurvPATH [56] (WSI Only) | 91.2 $\pm$ 1.5 | 72.8 $\pm$ 3.2 | 61.9 $\pm$ 2.7 | 70.5 $\pm$ 2.8 |
| HealNET (WSI Only) | 92.3 $\pm$ 1.3 | 72.9 $\pm$ 3.2 | 61.9 $\pm$ 2.1 | 71.1 $\pm$ 2.3 |
| **CustOmics (WSI Only)** | **92.5 $\pm$ 1.2** | **73.2 $\pm$ 3.1** | **62.2 $\pm$ 2.0** | **71.4 $\pm$ 2.2** |
| DeepSets [47] (WSI + Multi-Omics) | 96.7 $\pm$ 1.5 | 84.9 $\pm$ 2.0 | 58.6 $\pm$ 2.2 | 67.1 $\pm$ 2.7 |
| AttnMIL [22] (WSI + Multi-Omics) | 97.1 $\pm$ 1.2 | 86.6 $\pm$ 2.1 | 60.0 $\pm$ 2.5 | 69.4 $\pm$ 2.7 |
| DeepAttnMISL [45] (WSI + Multi-Omics) | 97.8 $\pm$ 1.1 | 88.3 $\pm$ 2.7 | 65.4 $\pm$ 2.7 | 66.6 $\pm$ 2.2 |
| MCAT [8] (WSI + Multi-Omics) | 98.9 $\pm$ 1.1 | 95.4 $\pm$ 2.0 | 93.3 $\pm$ 1.2 | 88.7 $\pm$ 2.3 |
| SurvPATH [56] (WSI + Multi-Omics) | 99.1 $\pm$ 1.0 | 96.5 $\pm$ 1.8 | 93.8 $\pm$ 1.1 | 92.5 $\pm$ 2.3 |
| HealNET (WSI + Multi-Omics) | 99.3 $\pm$ 1.0 | 98.5 $\pm$ 1.2 | 94.5 $\pm$ 1.1 | 96.1 $\pm$ 2.5 |
| **CustOmics (WSI + Multi-Omics)** | **99.5 $\pm$ 0.9** | **98.7 $\pm$ 1.1** | **94.7 $\pm$ 1.0** | **96.3 $\pm$ 2.4** |

**Table 2. Survival prediction performances.** Comparison of the discrimination of the models for the 8 TCGA cohorts, determined by the Concordance Index (C-index %) $\pm$ the standard deviation across the folds.

| Methods | BLCA | BRCA | COAD | GBMLGG | KIRC | LUAD | STAD | UCEC |
|---|---|---|---|---|---|---|---|---|
| SNN [27] (Multi-Omics) | 54.6 $\pm$ 2.7 | 47.5 $\pm$ 4.9 | 50.8 $\pm$ 3.4 | 60.5 $\pm$ 3.1 | 59.0 $\pm$ 5.7 | 54.2 $\pm$ 5.4 | 51.1 $\pm$ 4.6 | 49.6 $\pm$ 8.3 |
| HealNET (Multi-Omics) | 63.8 $\pm$ 1.6 | 63.0 $\pm$ 1.9 | 58.3 $\pm$ 1.9 | 78.0 $\pm$ 1.7 | 66.2 $\pm$ 2.4 | 63.4 $\pm$ 1.3 | 55.3 $\pm$ 1.5 | 68.2 $\pm$ 2.9 |
| **CustOmics (Multi-Omics)** | **64.1 $\pm$ 1.5** | **63.4 $\pm$ 1.8** | **58.6 $\pm$ 1.8** | **78.4 $\pm$ 1.6** | **66.5 $\pm$ 2.3** | **63.7 $\pm$ 1.2** | **55.6 $\pm$ 1.4** | **68.5 $\pm$ 2.8** |
| DeepSets [47] (WSI Only) | 50.6 $\pm$ 5.1 | 50.1 $\pm$ 6.5 | 50.2 $\pm$ 3.9 | 50.9 $\pm$ 3.5 | 49.6 $\pm$ 5.4 | 50.7 $\pm$ 7.5 | 50.0 $\pm$ 7.5 | 50.3 $\pm$ 7.5 |
| AttnMIL [22] (WSI Only) | 54.8 $\pm$ 4.8 | 57.0 $\pm$ 5.7 | 59.6 $\pm$ 3.9 | 79.0 $\pm$ 3.2 | 56.9 $\pm$ 6.1 | 56.3 $\pm$ 6.8 | 57.8 $\pm$ 6.1 | 63. $\pm$ 7. |
| DeepAttnMISL [45] (WSI Only) | 50.3 $\pm$ 5.5 | 52.1 $\pm$ 5.9 | 53.1 $\pm$ 3.3 | 73.8 $\pm$ 3.9 | 56.9 $\pm$ 6.6 | 55.0 $\pm$ 6.1 | 58.8 $\pm$ 5.3 | 60.0 $\pm$ 7.2 |
| MCAT [8] (WSI Only) | 55.5 $\pm$ 3.2 | 57.1 $\pm$ 5.6 | **59.9 $\pm$ 2.5** | **79.4 $\pm$ 2.0** | 56.0 $\pm$ 3.4 | 55.0 $\pm$ 6.1 | 57.7 $\pm$ 3.9 | 63.4 $\pm$ 6.7 |
| SurvPATH [56] (WSI Only) | 56.6 $\pm$ 3.4 | 58.1 $\pm$ 4.8 | 59.2 $\pm$ 2.3 | 79.0 $\pm$ 2.9 | 59.2 $\pm$ 2.7 | 58.5 $\pm$ 5.8 | 58.9 $\pm$ 3.1 | 65.6 $\pm$ 4.5 |
| HealNET (WSI Only) | 56.4 $\pm$ 4.0 | 59.0 $\pm$ 3.8 | 58.2 $\pm$ 2.2 | 78.3 $\pm$ 2.1 | 61.0 $\pm$ 2.3 | 59.7 $\pm$ 5.6 | 59.3 $\pm$ 2.4 | 67.6 $\pm$ 2.3 |
| **CustOmics (WSI Only)** | **56.7 $\pm$ 3.9** | **59.4 $\pm$ 3.7** | 58.5 $\pm$ 2.1 | 78.7 $\pm$ 2.0 | **61.3 $\pm$ 2.2** | **60.0 $\pm$ 5.5** | **59.6 $\pm$ 2.3** | 67.9 $\pm$ 2.2 |
| DeepSets [47] (WSI + Multi-Omics) | 59.6 $\pm$ 4.7 | 52.1 $\pm$ 7.1 | 61.4 $\pm$ 3.6 | 81.8 $\pm$ 3.3 | 54.2 $\pm$ 5.4 | 56.8 $\pm$ 7.3 | 52.9 $\pm$ 5.8 | 59.0 $\pm$ 7.4 |
| AttnMIL [22] (WSI + Multi-Omics) | 57.4 $\pm$ 5.3 | 54.8 $\pm$ 6.4 | 61.9 $\pm$ 3.5 | 81.3 $\pm$ 3.0 | 62.0 $\pm$ 6.2 | 58.5 $\pm$ 6.7 | 54.0 $\pm$ 6.2 | 56.9 $\pm$ 6.1 |
| DeepAttnMISL [45] (WSI + Multi-Omics) | 58.5 $\pm$ 5.4 | 58.0 $\pm$ 7.6 | 61.0 $\pm$ 3.2 | 81.4 $\pm$ 3.3 | 60.7 $\pm$ 7.1 | 55.0 $\pm$ 6.1 | 53.8 $\pm$ 5.7 | 59.1 $\pm$ 6.6 |
| MCAT [8] (WSI + Multi-Omics) | 62.4 $\pm$ 3.6 | 58.3 $\pm$ 5.5 | 62.8 $\pm$ 2.9 | 82.5 $\pm$ 2.3 | 66.5 $\pm$ 3.9 | 62.5 $\pm$ 4.5 | 56.2 $\pm$ 3.1 | 62.2 $\pm$ 2.7 |
| SurvPATH [56] (WSI + Multi-Omics) | 62.5 $\pm$ 3.1 | 65.5 $\pm$ 4.6 | 63.7 $\pm$ 2.2 | 83.4 $\pm$ 2.3 | 67.0 $\pm$ 3.3 | 63.5 $\pm$ 4.0 | 57.7 $\pm$ 2.3 | 65.0 $\pm$ 2.5 |
| HealNET (WSI + Multi-Omics) | 67.0 $\pm$ 2.6 | 64.9 $\pm$ 3.7 | 64.3 $\pm$ 2.3 | 83.9 $\pm$ 2.4 | 68.0 $\pm$ 2.2 | 64.6 $\pm$ 3.8 | 57.8 $\pm$ 1.6 | 67.8 $\pm$ 2.3 |
| **CustOmics (WSI + Multi-Omics)** | **67.2 $\pm$ 2.5** | **65.2 $\pm$ 3.6** | **64.5 $\pm$ 2.2** | **84.2 $\pm$ 2.3** | **68.2 $\pm$ 2.1** | **64.9 $\pm$ 3.7** | **58.0 $\pm$ 1.5** | **68.0 $\pm$ 2.2** |

To underscore the method's robustness concerning sample size, we evaluated smaller datasets, focusing on predicting molecular subtypes within three specific TCGA cohorts: TCGA-BRCA, TCGA-COAD, and TCGA-STAD. Across all classification tasks (as detailed in Table 1), Multimodal CustOmics consistently outperformed other comparable methods. Notably, in multi-omics integration, the mixed-integration VAE within Multimodal CustOmics demonstrated superior performance compared to the SNN utilized by other methods. An additional assessment showcased the significance of multi-omics integration, delineated in S4 Table, emphasizing the impact on performance when replacing the VAE encoder in Multimodal CustOmics with a standard SNN.

Further exploration into the integration strategies revealed the advantage of Multimodal CustOmics in exploiting diverse molecular data sources. Comparative analyses in S3 Table

demonstrated that while SNN performed best with RNAseq alone, Multimodal CustOmics enhanced performance when integrating RNAseq with CNV and methylation data. This divergence in integration strategies suggests Multimodal CustOmics' capacity to augment predictive power and unveil novel interactions among disparate data sources.

Regarding the survival outcome prediction across eight TCGA cohorts, Multimodal CustOmics' displayed consistently encouraging predictive capabilities compared to the alternative methods. Notably, in survival analysis based solely on WSI, Multimodal CustOmics exhibited comparatively weaker performance in specific cohorts than the transformer architecture employed in MCAT. An ablation study (detailed in S4 Table) replaced Multimodal CustOmics' hypergraph encoding for WSI embeddings with a visual transformer. While the transformer architecture demonstrated superior performance in analyzing individual WSI patches, the hypergraph-based approach achieved significantly better results when modeling region-wide tissue relationships that were then integrated with other modalities. This performance disparity highlights that although hypergraphs may not optimize patch-level feature extraction, their ability to encode relationships across broader tissue regions makes them particularly effective for capturing histological patterns at multiple scales. The limitations of the transformer's pairwise attention mechanism become apparent when modeling extended spatial relationships, where hypergraph-encoded region-wide representations provide a more comprehensive view of tissue architecture. Furthermore, as shown in S7 Table, hypergraph architectures are inherently more parameter-efficient, requiring significantly fewer weights compared to transformer models while maintaining effective region-wide feature learning. These findings reinforce our architectural choice for histopathology encoding, suggesting that optimizing for region-wide tissue relationships with a lighter computational footprint, rather than local patch-level features alone, leads to more effective representation learning that better complements multimodal integration.

In survival tasks, particularly in multi-omics scenarios (without WSI), Multimodal CustOmics obtained better prediction performances, especially as other methods like SNN showed concordance indices approaching randomness (Table 2). Of note, Multimodal CustOmics had a lower standard deviation across folds than other state-of-the-art approaches.

While the primary analysis was conducted using Cancer Hallmarks, we also performed the same analysis using multiple gene sets (Oncologic Signatures [58] and ReacTOME [57]). The results displayed in S5 and S6 Tables show similar performance to those obtained using Hallmarks. The change, however, may occur in the interpretability results that may vary depending on which set of pathways is chosen, changing the biological focus given to the model.

## 4.2. Multi-level interpretability: classification

Multimodal CustOmics can conduct pathway enrichment analysis across multiple tasks. Fig 2 presents interpretability findings concerning the PAM50 subtype classification within the TCGA-BRCA dataset. The objective is to elucidate the determinants driving the discrimination of specific subtypes, notably the Her2 subtype, within a multimodal context.

The initial layer of interpretability operates at the gene level, utilizing a Multi-Omics Pathway Enrichment Score (MPES) and conducting Gene Set Variation analysis using normalized gene importance scores (details in the methods section). Fig 2B delineates essential pathways in Her2 subtype discrimination, notably highlighting the significance of estrogen response and KRAS signaling down hallmarks. The interrelation between the Her2 subtype and estrogen response has been extensively investigated [33], emphasizing their coexpression's multifaceted impact on breast carcinogenesis, invasive behavior, and cellular growth.

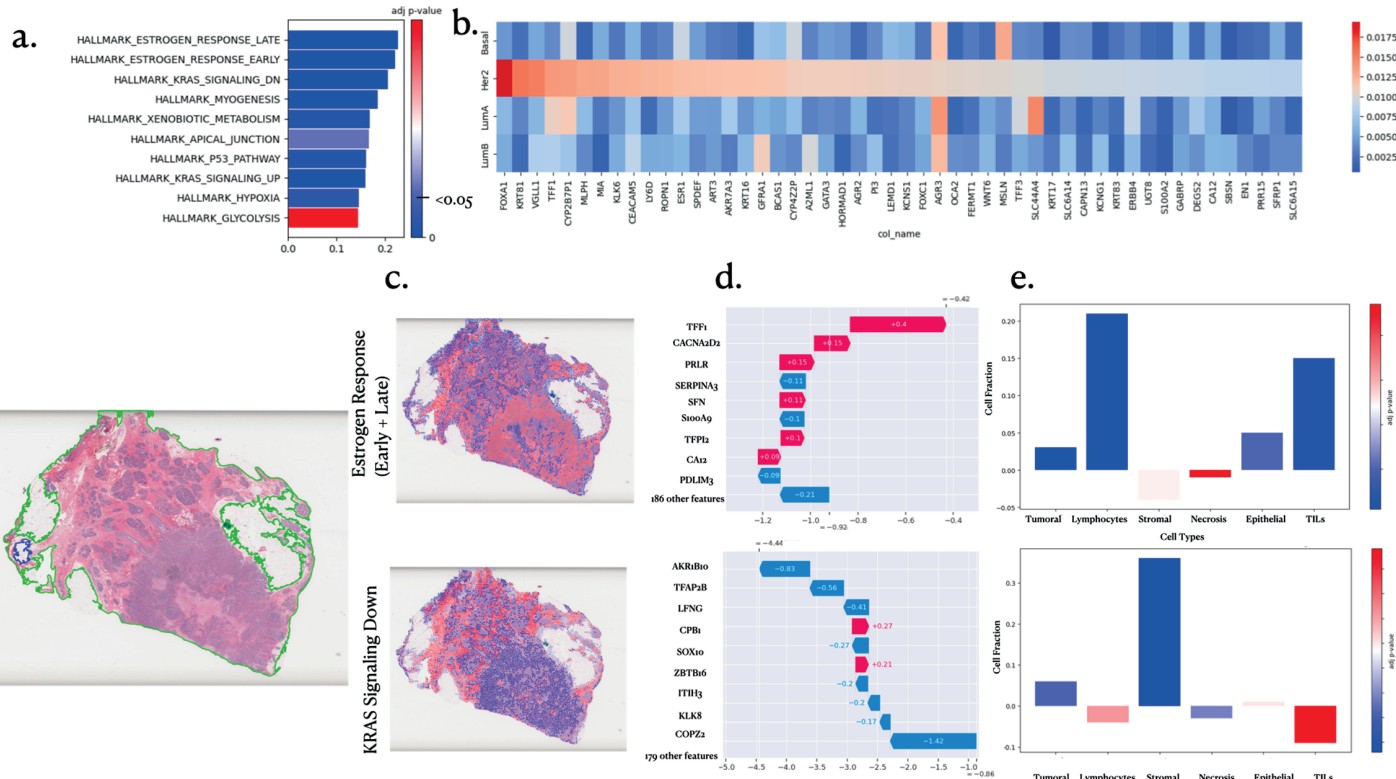

**Fig 2. PAM50 interpretability analysis. a.** Pathway enrichment scores and associated p-values from the gene set variation analysis. The coloured contour on the slide on the left is here to show good tissue segmentation. **b.** SHAP values for the most influential genes affecting the stratification of the Her2 subtype and their impact on other subtypes. **c.** Spatial Enrichment Analysis for the top 2 pathways and their key genes. **d.** Gene importance within the considered pathways. **e.** Differential Analysis of cell distribution between high and low attention regions.

Further exploration into gene-level importance is depicted in Fig 2C, spotlighting the predominant genes responsible for discriminating the Her2 subtype and contrasting their importance across other subtypes. Notably, the FOXA1 gene emerges with substantial importance, aligning with its suggested role as a transcription factor for Her2, as indicated in Cruz et al. [11].

Beyond multi-omics pathway enrichment analysis, Multimodal CustOmics extends interpretability to encompass multimodal enrichment, revealing spatial interactions within histopathology slides that correlate with specific pathways for discriminating Her2 subtypes. Fig 2D illustrates such interpretability outcomes for the estrogen response and KRAS pathways. Different cell populations within high-importance regions for each pathway are described to help biological interpretation. Notably, regions associated with the KRAS pathway exhibit increased proportions of stromal cells, suggesting potential regulation of tumor cell signaling via these stromal cells [42]. Conversely, the estrogen response demonstrates a strong interaction with regions featuring elevated densities of Tumor-Infiltrating Lymphocytes (TILs) and lymphocytes, corroborated by multiple sources [12,29]. This interaction holds particular significance when stratifying between ER- and ER+ patients.

## 4.3. Multi-level interpretability: survival

In a similar vein, survival analysis also lead to interpretable results. Fig 3 delineates the varying degrees of enrichment analysis for predicting survival outcomes within the TCGA Pan-cancer dataset.

Specifically, Fig 3B underscores the predominant influence of the inflammatory response pathway on survival outputs, a finding consistent with existing literature [51].

Demonstrating the relevance of employing multimodal integration in pan-cancer survival analysis, Fig 3C illustrates the impact of incorporating multiple data sources on stratifying low and high-risk patients, as evidenced by Kaplan-Meier curves and their corresponding log-rank p-values.

Further investigation into the effect of essential pathways is portrayed in Fig 3D, showcasing the significance of interactions between the two most crucial pathways. Notably, heightened importance is observed within the inflammatory response pathway, characterized by increased lymphocyte densities and Tumor-Infiltrating Lymphocytes (TILs). In contrast, the epithelial-mesenchymal transition pathway manifests greater densities of stromal cells.

Delving deeper into the Epithelial-Mesenchymal Transition pathway, the primary influential gene appears to be FBN2, renowned for its inhibition of cancer cell invasion and migration, as stated in Mahdizadehi et al. [32], thereby explaining its inclination toward lower risk outcomes.

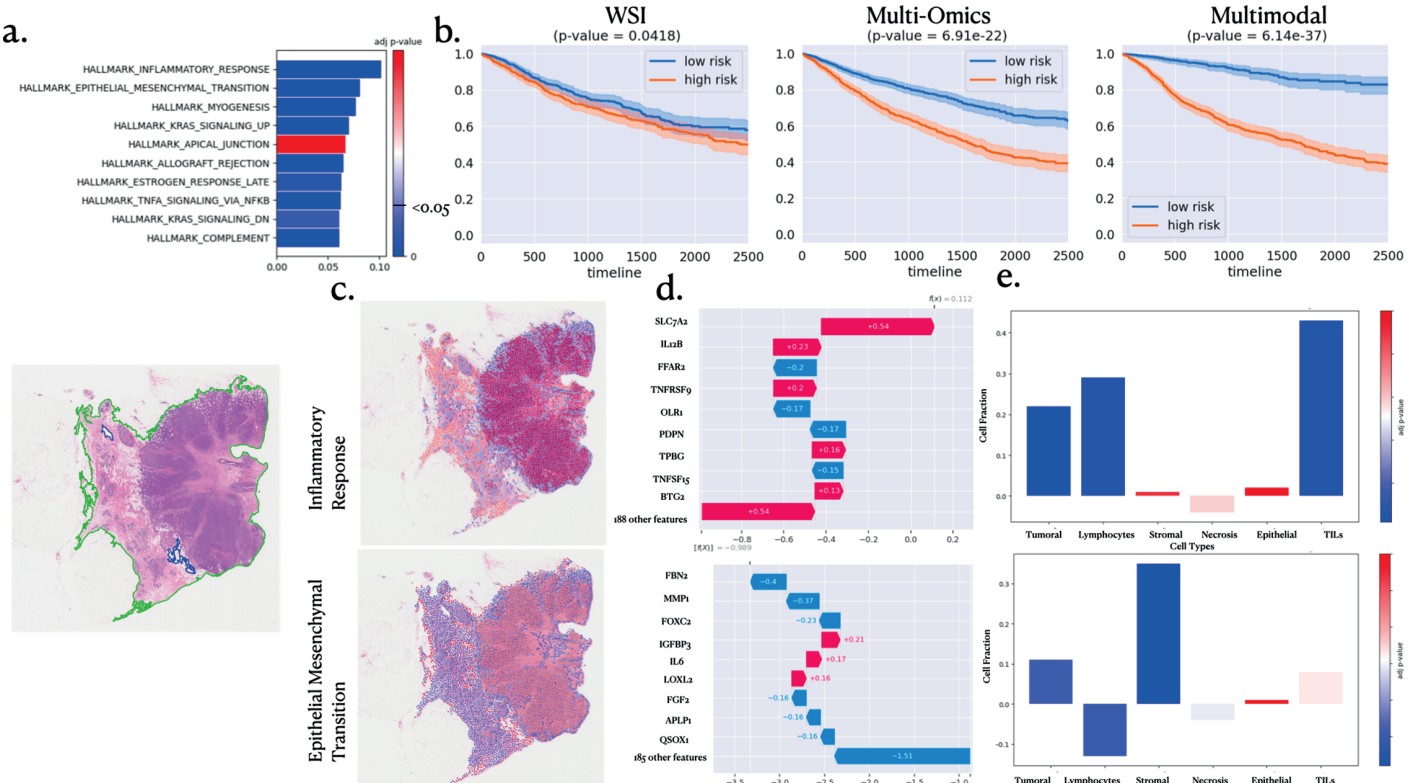

**Fig 3. Interpretability analysis for pan-cancer survival outcome prediction task. a.** Pathway enrichment analysis. The coloured contour on the slide on the left is here to show good tissue segmentation. **b.** Kaplan Meier curve associated with the survival outcome prediction task, showing the high and low risk for death event stratification with a computed log-rank p-value. **c.** Spatial Enrichment Analysis for the top 2 pathways and their most important genes. **d.** Gene importance within the considered pathways. **e.** Differential Analysis of cell distribution between high and low attention regions.

## 4.4. Multi-level interpretability validation

To ensure the reliability of interpretability findings by Multimodal CustOmics, we validated using the IALT data set, focusing on lung cancer. We undertook a comparative analysis of survival analysis interpretability results between the IALT dataset and a combined dataset of TCGA-LUAD/TCGA-LUSC for Lung Adenocarcinoma and Squamous Cell Carcinoma.

The pathway enrichment analysis, illustrated in Fig 4, revealed concordant outcomes between the TCGA and IALT datasets, identifying KRAS signaling down as the foremost pathway. This finding aligns with established literature [41], which underscores the prevalence of oncogenic KRAS mutations in approximately 25% of lung adenocarcinoma cases, thus representing a pivotal focus in current drug development.

Subsequently, we explored the interactions between the KRAS signaling-down pathway and spatial regions within histopathology images. Fig 4B depicts that regions of high attention in both datasets exhibit similar distributions of cell types, indicating the robustness of our method across distinct datasets of the same cancer type. Notably, this distribution underscores the association between the KRAS pathway and tumor-infiltrating lymphocytes (TILs) heightened densities and marginally increased tumoral cell counts in predicting survival outcomes. This consistent association echoes previous findings in [28], highlighting a solid correlation between KRAS mutation status and tumor immunity-related characteristics, notably CD8+ TILs.

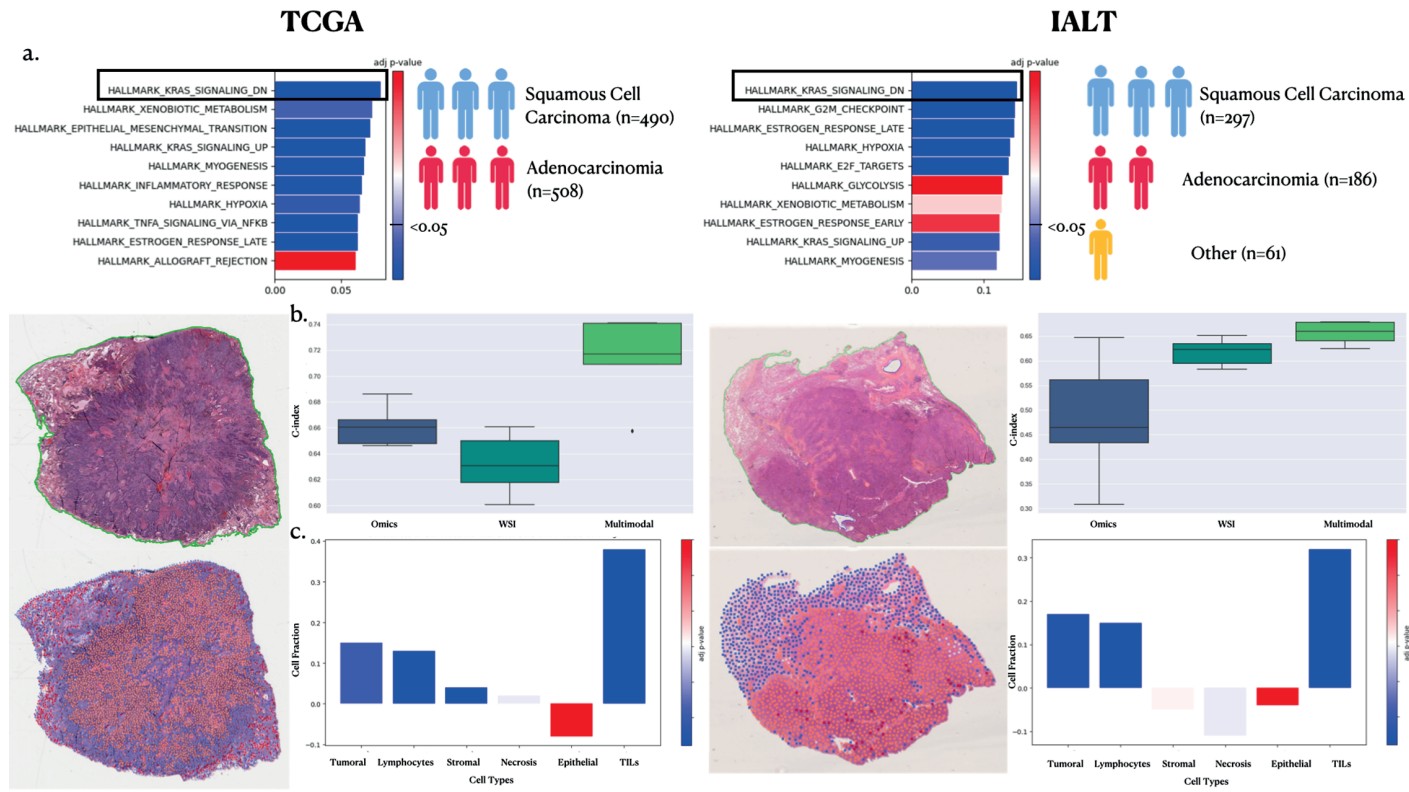

**Fig 4. Comparison of interpretations between the TCGA-LUAD/TCGA-LUSC datasets and IALT regarding the distinction between high and low survival risk.** **a.** Pathway Enrichment analysis highlights the task's top pathways. **b.** Survival Outcome Prediction Performances using C-index on Omics, WSI and Multimodal setup. **c.** Differential Analysis of cell distribution between high and low attention regions.

## 5. Discussion

The Multimodal CustOmics framework is a comprehensive toolbox to bridge prediction and interpretation within biological systems across multiple levels: genes, pathways, and spatial configurations. This multifaceted system generates three distinct interpretability scores concurrent with predictions, unraveling the biological knowledge underlying model outcomes. Empirical assessments underscore Multimodal CustOmics' robust predictive capabilities, outperforming state-of-the-art methodologies in integrating multi-omics and histopathology data across eight diverse datasets. However, despite its efficiency with smaller datasets in this study, Multimodal CustOmics' reliance on deep learning methodologies might restrict efficiency when confronted with limited training data availability.

Furthermore, Multimodal CustOmics stands out for its interpretability, facilitating a broader spectrum of analyses and enriching understanding across diverse biological modalities. Notably, although this study centered on three omics data types, Multimodal CustOmics exhibits versatility in seamlessly integrating varied omics data without necessitating framework alterations. This adaptability originates from an initial phase that independently trains on each source, serving as a normalization layer for heterogeneous sources.

The strategic segmentation of inputs into interpretable entities for spatial and molecular data mitigates challenges arising from the high dimensionality of whole slide images and multi-omics datasets. This partitioning augments interpretability and broadens the method's applicability to uncharted pathways or spatial regions beyond this study's scope. Multimodal CustOmics places significant emphasis on expansive interpretability functionalities. This aims to unveil predominant biological functions steering specific predictions across diverse data sources and scales. This comprehensive approach fosters collaboration between biologists and computational pathologists, offering a framework for in-depth analyses through enrichment analysis for omics and spatial data. The method extracts coherent insights from diverse data sources, unveiling a panoramic view of interconnected biological processes influencing the outcome of interest. By integrating omics and spatial data within enrichment analysis, Multimodal CustOmics enables a deeper understanding of the interplay between molecular information and spatial contexts, enriching investigative pathways for researchers in the field.

In addition, our Mixture-of-Experts (MoE) approach offers distinct advantages over traditional attention-based interpretability methods. By explicitly modeling the hierarchical nature of biological systems through specialized expert networks for different pathways and spatial regions, our approach achieves more stable and reliable interpretations of model decisions. As demonstrated in S1B Fig, our systematic stability analysis across the three TCGA subtype classification tasks reveals that the MoE-based architecture produces notably more consistent pathway importance scores compared to attention mechanisms. The rank correlation between weights from different training folds was substantially higher for MoE ($\rho \sim 0.75$ – $0.82$) than attention mechanisms ($\rho \sim 0.45$ – $0.65$), with this difference becoming more pronounced in smaller datasets like STAD. This enhanced stability is particularly crucial in cancer genomics, where sample sizes are often limited and reliable feature importance is essential for downstream analysis and clinical interpretation. The superior consistency of our MoE-based interpretations suggests that the identified important pathways are more robust to variations in training data, providing a more dependable foundation for biological insights.

Regarding the missing modality strategy, our analysis revealed important insights into the model's behavior under different missingness patterns. While our approach primarily operates under a Missing at Random (MAR) assumption, we investigated its robustness to Missing Not at Random (MNAR) scenarios through systematic experiments. By selectively removing

RNAseq data from Lung Cancer samples during training while preserving CNV and methylation data, we discovered that the model can inadvertently learn to use missingness patterns as predictive shortcuts. This was evidenced by elevated Lung False Positive Classification Rates (LFPR) at low dropout rates during testing, as shown in S1A Fig. The figure illustrates the evolution of LFPR across different dropout rates, with optimal performance observed at moderate dropout rates (around 40%), suggesting that introducing controlled missingness during training forces the model to rely more heavily on genuine biological signals from available modalities rather than missingness patterns. However, the persistence of elevated LFPR compared to MAR baseline performance indicates that completely mitigating learned biases from systematic missingness remains challenging. These findings underscore the importance of careful consideration of missing data patterns in multimodal integration approaches and suggest that strategic data dropout during training may help improve model robustness.

Despite its potential and performance, our method has a few noteworthy limitations. Firstly, while this study successfully delineates interactions between omics and histopathology data, it lacks a method to effectively discern how important individual contributions of each omic source are to a patient's molecular profile. It prevents a comprehensive understanding of each omic source's distinct impact on the molecular landscape. Secondly, the link established between the generated representation and phenotype data, beyond mere predictive labels, solely relies on the conditioning of the latent space. While this conditioning methodology effectively incorporates phenotypical signals into the multimodal representation, it lacks a mechanism to unveil interactions between different modalities and diverse clinical variables explicitly. This omission presents an avenue for future development, potentially enhancing the interpretability of multimodal interactions and their associations with clinical factors.

## 6. Conclusion

In this study, we introduced Multimodal CustOmics, a novel deep-learning framework that effectively integrates histopathology images and multi-omics data to enhance cancer characterization and patient outcome predictions. Multimodal CustOmics demonstrates superior performance in both classification and survival tasks, outperforming existing state-of-the-art methods. Our model provides robust and interpretable results, enabling the extraction of meaningful insights at gene, pathway, and spatial levels. This multi-level interpretability is critical for understanding the complex biological interactions within the tumor microenvironment and advancing precision medicine. The robustness of Multimodal CustOmics to missing data, through a multiple-instance variational autoencoder with modality dropout, ensures its practical applicability in clinical settings. Our extensive evaluations using TCGA datasets and validation cohorts underscore the potential of Multimodal CustOmics to improve diagnostic accuracy, clinical outcome predictions and thus therapeutic decision-making, paving the way for more personalized and effective cancer treatments. This Multimodal CustOmics framework is publicly available in Github.

## Supporting information

**S1 Table. TCGA dataset description.** Description of the TCGA datasets used in this study for classification and survival tasks. We show the number of patients available for each modality along with the censoring rate of the cohort.
(XLSX)

**S2 Table. IALT dataset description.** Overview of the IALT cohort with details about the different clinical and molecular features involved.
(XLSX)

**S3 Table. Modality combinations.** Performance comparison between multiple combination of modalities for Multimodal CustOmics for the Pancancer classification task. The evaluation is done using the Area Under ROC-curve (AUC).
(XLSX)

**S4 Table. Ablation study.** Performance comparison between Multimodal CustOmics and the state of the art for classification tasks by replacing different model instances: **a. Multi-Omics Ablation:** CustOmics A1 replaces the multi-omics VAE with an SNN. **b. Hypergraph Ablation:** CustOmics A2 replaces the hypergraph encoder with a visual transformer and CustOmics A3 replaces the hypergraph represnetation with a regular graph embedding. **c. Downstream Network Ablation:** CustOmics A4 replaces the hierarchical mixture-of-experts approach with a regular mixture of experts network, CustOmics A5 replaces it with a transformer classifier.
(XLSX)

**S5 Table. Multiple gene sets study classification.** Multimodal CustOmics' classification perfomances across all tasks for multiple gene sets.
(XLSX)

**S6 Table. Multiple gene sets study survival.** Multimodal CustOmics' survival perfomances across all tasks for multiple gene sets.
(XLSX)

**S7 Table. Comparison of learnable parameters in Transformer and Hypergraph models for WSI embedding.**
(XLSX)

**S1 Fig. Model robustness analysis and hyperparameter optimization. a.** Evolution of Lung False Positive Rate (LFPR) across different RNAseq dropout rates during testing, with baseline Missing at Random (MAR) performance (green dashed line) and 95% confidence intervals (shaded area). **b.** Rank correlation comparison between MoE and attention-based feature importance across three TCGA cancer types (BRCA, COAD, STAD) with error bars indicating standard deviation. **c.** Ablation studies showing model performance sensitivity to hypergraph construction parameters: similarity threshold (left) and kernel parameter ratio (right), with 95% confidence intervals (shaded areas).
(TIF)

**S1 Text. SHAP values.** Ideas and methodology behind SHAP values.
(PDF)

**S2 Text. Survival analysis.** Fundamentals of Survival Outcome Prediction Tasks.
(PDF)

**S3 Text. Mixture of experts.** Methodology behind the standard mixture-of-expert (MoE) framework.
(PDF)

## Author contributions

**Conceptualization:** Hakim Benkirane.

**Data curation:** David Planchard, Julien Adam, Ken Olaussen.

**Formal analysis:** Hakim Benkirane.

**Funding acquisition:** Stefan Michiels, Paul-Henry Cournède.

**Methodology:** Hakim Benkirane.

**Resources:** Paul-Henry Cournède.

**Software:** Hakim Benkirane.

**Supervision:** Maria Vakalopoulou, Stefan Michiels, Paul-Henry Cournède.

**Validation:** Hakim Benkirane, Ken Olaussen, Stefan Michiels, Paul-Henry Cournède.

**Visualization:** Hakim Benkirane.

**Writing – original draft:** Hakim Benkirane, Maria Vakalopoulou, Stefan Michiels, Paul-Henry Cournède.

**Writing – review & editing:** Hakim Benkirane, Stefan Michiels, Paul-Henry Cournède.

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
