## [Decision Letter · Decision Letter 0]

20 Dec 2024

PCOMPBIOL-D-24-01276

Multimodal CustOmics: A Unified and Interpretable Multi-Task Deep Learning Framework for Multimodal Integrative Data Analysis in Oncology

PLOS Computational Biology

Dear Dr. Cournède,

Thank you for submitting your manuscript to PLOS Computational Biology. After careful consideration, we feel that it has merit but does not fully meet PLOS Computational Biology's publication criteria as it currently stands. Therefore, we invite you to submit a revised version of the manuscript that addresses the points raised during the review process.

Please submit your revised manuscript within 30 days Feb 19 2025 11:59PM. If you will need more time than this to complete your revisions, please reply to this message or contact the journal office at ploscompbiol@plos.org. Please include the following items when submitting your revised manuscript:

We look forward to receiving your revised manuscript.

Kind regards,

Eric Gamazon

Academic Editor

PLOS Computational Biology

Pedro Mendes

Section Editor

PLOS Computational Biology

**Journal Requirements:**

3) Some material included in your submission may be copyrighted. According to PLOSu2019s copyright policy, authors who use figures or other material (e.g., graphics, clipart, maps) from another author or copyright holder must demonstrate or obtain permission to publish this material under the Creative Commons Attribution 4.0 International (CC BY 4.0) License used by PLOS journals. Please closely review the details of PLOSu2019s copyright requirements here: PLOS Licenses and Copyright. If you need to request permissions from a copyright holder, you may use PLOS's Copyright Content Permission form.

Potential Copyright Issues:

- Figure 4A; Please confirm whether you drew the images / clip-art within the figure panels by hand. If you did not draw the images, please provide a link to the source of the images or icons and their license / terms of use; or written permission from the copyright holder to publish the images or icons under our CC BY 4.0 license. Alternatively, you may replace the images with open source alternatives. See these open source resources you may use to replace images / clip-art:

4) Please ensure that the funders and grant numbers match between the Financial Disclosure field and the Funding Information tab in your submission form. Note that the funders must be provided in the same order in both places as well.

**Reviewers' comments:**

Reviewer's Responses to Questions

**Comments to the Authors:**

Reviewer #1: **Summary**

This paper proposes a deep learning method to jointly model multi-omics and histopathology slides. The model can handle incomplete data (e.g. patients might be missing different modalities) and aims to shed light on the activity of different pathways in different cancer types. The performance of the method is demonstrated on data from The Cancer Genome Atlas and International Adjuvant Lung Cancer Trial.

**Strengths**

* The proposed method is flexible: it can model histopathology slides together with multiple omics modalities. Using contrastive learning to extract features from different patch makes the problem more manageable, allowing to jointly extract information from whole slides. Overall, the model is interesting and original, but the manuscript could benefit from some clarifications (please see comments).

* The approach focuses on interpretability without compromising performance. The idea of partitioning a slide into multiple regions and omics data into multiple pathways is interesting. The intermediate region by pathway features could be useful for histopathologists to gain insights into cancer biology.

* Overall, the interpretability results are compelling. The method seems to recapitulate pathways and genes important for PAM50 classification and patient survival. The authors used a method to categorize cells into different classes including tumors, lymphocytes,... Using this approach, they could verify what the method pays attention to within the high-attention image patches, i.e. the model seems to pay attention to patches with high tumor cell density.

**Weaknesses**

- The motivation of using a hypergraph autoencoder is unclear. Why is this necessary? The ablation study in Table S4 shows that alternative encoders (e.g. based on a regular graph embedding, A3, or a hierarchical mixture of experts, A4) perform similar, if not better, than the hypergraph approach. Overall, the rationale behind this modelling decision could be improved, e.g. why is a hypergraph necessary? why is the hypergraph constructed based on morphological similarity versus spatial proximity?

- It is unclear how the model handles different modalities (i.e. under what missingness assumption it operates) and whether it is affected by artifacts. Can you demonstrate that the missingness pattern is not predictive of the downstream task? Is it possible to prove that, if certain conditions are fulfilled, the method will work as intended? (e.g. if the data is missing not at random at train time, can you guarantee that the method will generalize well at test time?). Some studies have found that JPEG quality can have a strong confounding effect on histopathological feature representation and tissue classification in the TCGA cohort (see Fu et al., "Pan-cancer computational histopathology reveals mutations, tumor composition and prognosis"). To what extent do these confounding factors influence the method's predictions?

- The benchmark could be improved to account for state-of-the-art methods that have recently been released. For example, a recently introduced state-of-the-art method (HEALNet) addresses the same multi-modality integration goal. Could the authors include this method in their benchmark? (see Hemke et al., "[HEALNet: Hybrid Multi-Modal Fusion for Heterogeneous Biomedical Data"). This and potentially other methods should be included in the benchmark to claim state-of-the-art performance.](https://arxiv.org/abs/2311.09115)

- The motivation and advantages of the proposed interpretability method are unclear. What is the advantage of this approach over existing interpretability methods that already account for the architecture of the predictive model? Can the authors show the advantages of the proposed interpretability approach over existing methods? For example, what is the advantage of weighting SHAP values by pathway enrichment scores (e.g. with respect to using plain SHAP values)?

**Questions and comments**

* L70: "The method outputs the importance score of each gene for each molecular source, both independently and in association with other modalities". To what extent can we trust these scores? Do you have any experiments showing that the method indeed recovers known biomarkers? What does "in association with other modalities" mean exactly?

* L74-76: "This score has also been extended to account for spatial correlations in the histopathology slide and reflects the importance of the interaction between spatial regions of the WSI and pathways". How has the score been extended to account for spatial correlations?

* Fig 1: How is the pathway aggregation carried out? Does this method require prior pathway information or is it purely data-driven?

* Fig 1, caption: "Phase 1 learns the weights of each region inside a pathway" -> did the authors intend to say "Phase 1 learns the weights of each pathway inside a region"?

* L96-97: How is the ResNet-18 trained via contrastive learning? Details seem to be missing.

* L106-L114: How are the patches with high similarities grouped into clusters. Did the authors used a threshold based on the path similarity provided by the kernel? If so, how is this threshold selected? How are $\lambda_h$ and $\lambda_p$ defined?

* L114-L117: How is $\mathcal{P}_i$ defined? Is the clustering algorithm applied to further group the patch groups from the previous step into K categories? Is the interpretation by the pathologist a necessary step to select relevant regions, or is this rather a byproduct of the methodology? (i.e. the identified regions are amenable to interpretation by histopathologists)

* L118-L121: Am I right in thinking that nodes in the hypergraph are "grouped patches" and that hyperedges group these patches whenever their morphological similarity exceeds a threshold? I recommend 1) clarifying these points in the text, 2) removing unnecessary symbols (e.g. are $G_i$, $V_i$, and $E_i$ used anywhere else in the text?), and 3) including a clear definition of each symbol (e.g. $p_j$ and $\mathcal{P}_i$).

*L129-131: How are the P gene sets determined? How are the variational autoencoder models trained?

* L131-133: "The encoding networks are designed to accept inputs from all omic sources about each gene set". How are these inputs represented? Could the authors clarify how the different omics modalities need to be represented? e.g. how did they align the CNV and methylation data with RNA-seq?

* L140-L141: Can the "bilinear fusion operator" be described in more detail? What are the advantages with respect to other standard modelling choices such as cross-attention?

* L150: What is the meaning of the sentence after "where (...)". Did the authors want to say that the weights must sum to 1?

* L155: What is the meaning of $(w_p)p$? There seem to be some notation issues.

*L171-L177: How are the modalities dropped out, exactly? In Section 2.1.2, it is mentioned that a VAE is used to encode pathway-level features using inputs from all omic sources. In what part of the architecture is the modality dropout performed?

* L199-L207: What is gene set variation analysis? What is the advantage of weighting SHAP values by the pathway enrichment scores? Shouldn't these pathway scores be already implicitly embedded in the original SHAP values, which depend on the architecture? What happens if a gene belongs to multiple pathways? In general, the advantages of such an approach are unclear at this point.

* L238-L242: Why is the multiphase training strategy required? Can the authors provide an ablation study showing the benefits of such approach?

* L257: A recently introduced state-of-the-art method (HEALNet) addresses the same multi-modality integration goal. Could the authors include this method in their benchmark? [HEALNet: Hybrid Multi-Modal Fusion for Heterogeneous Biomedical Data](https://arxiv.org/abs/2311.09115)

* L332: What is the meaning of "gene segmentation"?

* Figure 2: What do the different colors in the panel before 2c represent?. Figure 2c: How was the spatial enrichment analysis carry out, and what are the observations/conclusions for that panel? Figure 2e: how was the differential analysis carried out? My understanding is that these cell distributions were calculated based on the attention maps of Figure 2c, is this correct? How do these cell fractions align with our understanding of breast cancer subtyping?

* Figure 3: Same questions as above.

* Figure 4b: What do the box plots show? Could this be described in the caption?

Reviewer #2: In this work, the authors introduce a multimodal, deep learning-based framework for tumor type classification and survival outcome prediction. Of particular note is their approach’s novel implementation of multivariable dropout, which allows for robust predictive performance even in the absence of some modalities. The authors benchmark their approach, Multimodal CustOmics, on a pan-cancer dataset from TCGA, demonstrating that it consistently outperforms several previously published approaches at both classification and survival prediction tasks. Overall, the authors present an innovative multimodal framework that addresses key limitations of existing models and shows robust performance against current state-of-the-art approaches. To enhancer the manuscript prior to publication, I have identified the following points where further clarification or refinements are needed:

Major comments:

•In line 116, what is the specific clustering algorithm used to divide the patch representations into categories?

•In line 118, what are Vi and Ei in the defined hypergraph Gi? Xi is defined in line 99, but I don’t see definitions for the other two variables.

•In lines 128-129, could the authors provide more detail regarding how these gene sets are defined?

•In the “4.1 Prediction Results” section starting on line 285, it would be helpful to note some of the specific performance metrics in the text.

•In Figures 2 and 3, the histology slide image on the far left should be labeled/annotated so that the reader can easily identify what is being depicted. The same should be done for the histology images in Figure 4.

•In Figures 2e, 3e, and 4c, the adjusted p-value scale should be labeled with numeric values.

•In the supplement, there is nothing listed under the “1 Supplementary Tables” header. Instead, all supplementary text and tables are combined under the “2 Supplementary Texts” header.

•There is some inconsistency in the naming of supplementary tables in the supplement. For example, Tables S5 and S6 are titled “Table 5” and “Table 6.”

•The supplement includes a “Text S3,” but this is not mentioned in the body of the main manuscript or noted under “Supporting information.”

Minor comments:

•In the Figure 1 caption, “Multimodal Dropout” is labeled differently from the “Modality Dropout” header in the figure.

•In lines 96 and 149, “inputted” should be changed to “input.”

•In Table 1, it would be helpful to bold the CustOmics approaches in the Methods column, like they are in Table 2.

•In Figure 2d, there is a halo effect around the numbers at the top and bottom of each plot.

**Have the authors made all data and (if applicable) computational code underlying the findings in their manuscript fully available?**

Reviewer #1: Yes

Reviewer #2: Yes

PLOS authors have the option to publish the peer review history of their article (what does this mean?). If published, this will include your full peer review and any attached files.

Reviewer #1: No

Reviewer #2: No

**Figure resubmission:**
---

## [Editor Report · Decision Letter 1]

31 Mar 2025

Dear Prof.Dr. Cournède,

We are pleased to inform you that your manuscript 'Multimodal CustOmics: A Unified and Interpretable Multi-Task Deep Learning Framework for Multimodal Integrative Data Analysis in Oncology' has been provisionally accepted for publication in PLOS Computational Biology.

Best regards,

Pedro Mendes, PhD

Section Editor

PLOS Computational Biology

---

## [Editor Report · Acceptance letter]

PCOMPBIOL-D-24-01276R1

Multimodal CustOmics: A Unified and Interpretable Multi-Task Deep Learning Framework for Multimodal Integrative Data Analysis in Oncology

Dear Dr Cournède,

I am pleased to inform you that your manuscript has been formally accepted for publication in PLOS Computational Biology. Your manuscript is now with our production department and you will be notified of the publication date in due course.

With kind regards,

Anita Estes
